# Vaccinating Children against SARS-CoV-2: A Literature Review and Survey of International Experts to Assess Safety, Efficacy and Perceptions of Vaccine Use in Children

**DOI:** 10.3390/vaccines11010078

**Published:** 2022-12-29

**Authors:** Lauren Hookham, Hillary C. Lee, Divya A. Patel, Mariana Coelho, Norberto Giglio, Kirsty Le Doare, Pia S. Pannaraj

**Affiliations:** 1Institute for Infection and Immunity, St. George’s University of London, London SW17 0RE, UK; 2Children’s Hospital Los Angeles, Los Angeles, CA 90027, USA; 3Independent Researcher, Buenos Aires Ciudad 1425, Argentina; 4Children’s Hospital Ricardo Gutiérrez, Buenos Aires Ciudad 1425, Argentina

**Keywords:** COVID-19 vaccines, SARS-CoV-2, children, adolescents, pediatrics

## Abstract

Introduction: The balance of risks and benefits of COVID-19 vaccination in children is more complex than in adults with limited paediatric data resulting in no global consensus on whether all healthy children should be vaccinated. We sought to assess the safety, efficacy, and effectiveness of childhood vaccination against SARS-CoV-2, as well as better understanding perceptions of vaccination in parents and vaccine experts. Methods: We performed a literature review for COVID-19 vaccine safety, efficacy, effectiveness, and perceptions. We searched international safety databases for safety data and developed an electronic survey to elicit country-specific COVID-19 immunisation data, including vaccine regulations, policies, rates, and public attitudes solicited from vaccine experts. Results: Nine studies were included in the final safety analysis. Local reactions were frequently reported across all studies and vaccine types. Adverse events reported to surveillance systems tended to be non-serious, and commonly included injection site reactions and dizziness. Twenty-three studies reported immunogenicity, efficacy, and effectiveness data. There were nine randomised control trials of six different vaccine types, which showed seroconversion of neutralising antibodies in vaccinated children ranging from 88% to 100%. The vaccine efficacy for Pfizer and Moderna vaccines ranged from 88% to 100%. There were 118 survey responses representing 55 different countries. Reported vaccination rates ranged from <1% to 98%. Most respondents described “mixed opinions” regarding paediatric vaccination policies in their country. By region, a more positive public attitude towards vaccination correlated with higher vaccination rates. Discussion: In this mixed-methods review, we have found evidence that vaccination against COVID-19 in children is safe, efficacious, and effective. Overall, the combined evidence from both the literature review and survey highlights the need for further data on both the safety and effectiveness of COVID-19 vaccinations in children.

## 1. Introduction

Though children are considered a population at “low risk” of severe COVID-19 illness and death [1,2,3], the medium- and long-term effects of the virus can be profound. SARS-CoV-2 is linked with two long-term conditions; multisystem inflammatory syndrome in children (MIS-C) [4,5,6], as well as post-COVID syndrome or “long COVID” [7,8,9], which encompasses a wide variety of clinical conditions. A recent meta-analysis revealed a prevalence of long COVID of 25.24% in children and adolescents [10], with a more recent study highlighting that adolescent girls may be at more increased risk of poor outcomes [11]. Hospitalisation rates in children in Europe have increased as a proportion of cases since January 2021 [12]. Furthermore, children have been identified as an important source of household and community transmission [13,14]. Critically, vaccination is a protective factor [15,16,17,18].

However, the balance of benefits of COVID-19 vaccination in children is more complex than in adults, with no global consensus on whether all healthy children should be vaccinated [19,20]. We sought to plug the information gap related to childhood vaccination against SARS-CoV-2 through a systematic review of the literature and a questionnaire sent to experts regarding childhood COVID-19 immunisation in order to describe and summarise existing vaccine safety, efficacy, and effectiveness data as well as current practices.

## 2. Methods

We searched EMBASE, PubMed, Cochrane, and Clinicaltrials.gov for studies describing COVID-19 vaccine safety, efficacy, and effectiveness in children ≤18 years of age. The search terms and strategy used for each individual area can be seen in Appendix A. Previous systematic reviews were reviewed, and all included trials were cross-checked against our own search. Databases were searched twice for data pertaining to safety, immunogenicity, efficacy, and effectiveness; initially on 6 December 2021 and then again on 20 April 2022 to capture as much data as possible. A final PubMed search was performed on 1 September 2022 to facilitate completeness of the data. 

We ran individual internet searches on 23 March 2022 and on 24 September 2022 for safety data of COVID-19 vaccination in children and adolescents from the following countries or areas: Australia, Ireland, Canada, the EU/EEA, USA, New Zealand, and the World Health Organisation (WHO) VigiAccess database. Despite extensive searches for national safety databases, we were unable to find any reporting data from low- and middle-income countries.

### 2.1. Eligibility Criteria

#### 2.1.1. Safety

Randomised control trials of all vaccines for SARS-CoV-2, in which a vaccine that presented safety data was given to children (<18 years old), were eligible for inclusion regardless of vaccine type or dosage. 

#### 2.1.2. Immunogenicity and Efficacy

All primary vaccine trials including individuals ≤ 18 years old that presented immunogenicity and/or efficacy data were included. 

#### 2.1.3. Vaccine Effectiveness (VE)

All studies evaluating effectiveness of vaccine against infection, hospitalisation, or severe infection or ICU admission were included.

### 2.2. Exclusion Criteria

Studies that did not include original data were excluded. Articles were excluded if they could not be translated by the authors into English. 

### 2.3. Selection and Data Collection Process 

Search outputs were imported into Rayaan.ai (https://rayyan.ai/, accessed on 3 December 2021). Safety data were reviewed by KLD and LH. Immunogenicity, efficacy, and effectiveness data were reviewed by PSP, HCL, and DAP. LH screened all abstracts to exclude duplicates. Discrepancies were resolved by review by a third author (KLD, NG, or PSP). If a study was identified with limited data, study authors were contacted to seek additional results. Authors were contacted again if there was no response after 2 weeks. 

### 2.4. Data Items 

Data extracted included: author, participant age, study type, intervention measures (vaccine type, number of doses, etc.), period of trial (as a rough analogue to predominant circulating SARS-CoV-2 strain), and trial location. For studies included in the safety review, adverse events (including definitions, solicited, unsolicited, and serious) were also obtained. Given the heterogeneity of reported adverse effects, all of those listed in the primary paper and any supplemental material were extracted to allow comparison between studies. Data not presented with a numerical count or clear percentage (i.e., a bar chart with no data figures elsewhere in the text) were excluded.

### 2.5. Effect Measures 

#### 2.5.1. Safety

We reported the percentage of participants (vaccine or placebo group) reporting an adverse effect (solicited, unsolicited and reactogenicity). In reviews of vaccine safety databases or vaccine safety surveys, we presented the percentage of vaccine recipients who reported an adverse event.

#### 2.5.2. Immunogenicity and Efficacy

We reported one or more of the following: the percentage of vaccinated participants showing seroconversion of neutralising antibodies, the geometric mean titre (GMT) after completion of the primary vaccine series, or the GMT ratio comparing immune response in children and adolescents to that in the young adult population. For efficacy data, we reported the percent reduction of confirmed COVID-19 or symptomatic infection in vaccinated participants compared with unvaccinated participants in the clinical trials. 

#### 2.5.3. Vaccine Effectiveness

VE was measured as the percent reduction of COVID-19 infection, COVID-19-associated hospitalisation, or critical care intervention requirements in vaccinated individuals compared with unvaccinated individuals in real-world settings using case-control test-negative study designs.

### 2.6. Synthesis Methods 

If possible, data were pooled and analysis undertaken using the Review Manager tool Version 5.4.1, The Cochrane Collaboration, 2020. Odds ratios and 95% confidence intervals (CI) were calculated when data were available in at least 3 separate studies for outcomes in studies of the same vaccine using the Mantel–Haenszel method. 

For immunogenicity, efficacy, and effectiveness data, reported outcome metrics from the individual studies were utilised. The 95% CI reported by each study was included. No additional data analysis was performed by the authors. 

### 2.7. Survey of Vaccine Experts 

We developed an electronic survey to elicit country-specific COVID-19 immunisation data, including vaccination policies, vaccines types being given to children, vaccination rates, and perceived public attitudes. The survey was sent to the Pregnancy, Maternal, Adolescent, and Child Health Working Group of the WHO and vaccine experts associated with the Advanced Course of Vaccinology (ADVAC) in 121 countries. The ADVAC network includes vaccination experts in academia, industry, government, and non-governmental agencies. The Qualtrics survey was sent with two reminders between April 2022 and May 2022. Data were collected through June 2022. For countries with more than one respondent, all individual responses were included in the analysis and average vaccination rates reported. 

#### Qualitative Data Analysis

An open-ended question was posed to respondents about their perceived major reasons for COVID-19 vaccine hesitancy in their country. The responses were reviewed to extract themes using directed content analysis [21]. The codebook was created after a primary search of the literature of the concepts and causes of vaccine hesitancy. Two coders (HL and DP) were used to establish trustworthiness of the qualitative findings [22]; they met to review and consolidate coding and reconcile any differences. Coding disputes were resolved through discussion with the co-author (PSP). A visual representation of the text that appeared more frequently was generated using Python 3.9.

### 2.8. Ethics

Approval was obtained from Children’s Hospital Los Angeles, CHLA-21-00316.

## 3. Results

PRISMA flow diagrams [23] detailing the results of the searches of the literature and selection process are presented in Appendix A. 

### 3.1. Safety Data

We found 13 studies that included children and reported safety data. Four studies provided no age-disaggregated data for safety for children, and instead data were presented within the entire study population. No response from study authors for data in children was received, and so these studies were excluded from the review. Subsequently, a total of nine RCTs were included in the analysis. These can be seen in Table 1. An additional 14 studies evaluated VE in the real-world setting after the emergency use authorization of the BNT162b2 (Pfizer-BioNTech) vaccine and the “China-made” COVID-19 vaccines. These can be seen in Table 2.

In total, over 10,000 children received a vaccine during trials conducted between August 2020 and August 2021. All trials involved children receiving at least two doses of vaccine, between 21 and 56 days apart. Eight of nine studies involved placebo, and all except one were blinded. The age of participants ranged from 6 months to 18 years old. Four RCTs reported evidence from trials with mRNA vaccines [24,26,31], three with inactivated virus particles [25,28,29,30], one of an adenovirus viral vector [25] and one of a DNA-based vaccine [32]. Studies were conducted in the USA, Canada, China, and India. There were no studies from Africa or South America. The most common local reaction amongst all vaccine types was injection-site pain, which was reported in up to 95% of vaccine participants and was more frequently described after the first dose of vaccine. 

Systemic reactions were reported in up to 68.5% of participants after dose 1 and 86.1% of participants after dose 2. The most common were fatigue and headache. Fever after the first dose was reported between 3 and 10% of participants, and between 1.1 and 24% of participants after dose 2. 

Studies varied in their presentation of unsolicited adverse events (AEs), including causality assessment. For studies that presented this data, unsolicited AEs related to the study vaccines were reported in 0.1 [29] to 7% [25] of participants. 

Across all studies, only two serious AEs (SAEs) were thought to be related to vaccination. One case of grade 3 allergic purpura after a second dose of BBIBP-CorV vaccine [30]. The second SAE occurred in a child who received an adenovirus vector vaccine and developed a gastrointestinal disorder requiring hospitalisation for six days [29]. Both children made a full recovery. No cases of thrombosis, myocarditis, MIS-C, or deaths were reported in any of the studies. 

Two participants withdrew secondary to AEs. Both participants received the BNT162b2 vaccine. One child, aged 12–15 years, had a temperature of >40 degrees which resolved after two days [24]. A second participant, aged 16–18 years, withdrew secondary to severe injection-site pain and headache which resolved after one day [24]. 

### 3.2. Safety Databases 

The results of 714,103 safety surveys undertaken by vaccinated children and adolescents aged 5 to 20 years (or their caregivers) in the USA [44] and Australia [45] were available for review (Appendix A). They had received either Pfizer (*n* = 704,007) or Moderna (*n* = 10,096) vaccines. In the USA, 63.4% of vaccinated children or adolescents reported local reactions, with injection-site pain being the most frequently described across all age groups [44]. Systemic reactions were reported in 48.9% of respondents, with fatigue, headache, and myalgia being the most common. Data from Australia showed a similar pattern, with local reactions and fatigue being reported more frequently in those who had received three doses [45]. 

Several national and multinational safety surveillance databases were also reviewed, including data from the USA [46], United Kingdom [47], Canada [48], New Zealand [49], Ireland [50], Australia [51], the European Union [52], and the World Health Organisation (WHO) [53], representing millions of vaccine doses. Adverse events reported to surveillance systems tended to be non-serious, and commonly included injection-site reactions and dizziness (see Appendix A). In the USA, where over 25 million children (ages 2–17) have completed a primary course of vaccination with the Pfizer vaccine, a total of 38,555 adverse events were reported to the Vaccine Adverse Event Reporting System (VAERS), with the minority being serious. A total of 1184 cases of myocarditis post vaccination have been reported (see Appendix A), with 1626 cases documented in the EU’s EudraVigilance platform [52]. In the USA and in the majority of other reports where vaccine type was available, cases were secondary to the mRNA vaccines. 

### 3.3. Results of Syntheses 

The mRNA vaccines showed an increased risk of injection-site pain after dose 1 and dose 2 of vaccine as compared with placebo (pooled odds ratio [OR] 10.59 [95% CI 9.67–11.59] after dose 1; OR 14.86 [95% CI 13.46–16.41] after dose 2) (Figure 1a). There was also an increased risk of erythema (dose 1 OR 7.36, 95% CI 5.7 6–9.40; dose 2 OR 10.58, 95% CI 8.27–13.55) and swelling (dose 1 OR 9.59, 95% CI 7.29–12.62; dose 2 OR 13.47, 95% CI 10.23–17.73). 

Systemic reactions were more common in those who received an mRNA vaccine as compared with placebo. The odds ratio of fever after dose 1 was 4.50 (95% CI 3.25–6.23) in those who received the vaccine. This risk trebled (OR 15.45, 95% CI 11.61–20.56) after dose 2. The risk of fatigue was similarly higher after dose 2 (OR 4.01, 95% CI 3.69–4.36) than dose 1 (OR 1.57, 95% CI 1.45–1.70) of the mRNA vaccine (Figure 1b). A similar trend was seen with headache, fatigue, and nausea and vomiting (Appendix A). 

### 3.4. Immunogenicity and Efficacy 

Nine randomised, placebo-controlled clinical trials provided immunogenicity and efficacy data. Of these, three provided paediatric data regarding the BNT162b2 (Pfizer-BioNTech) vaccine, 2 regarding mRNA-1273 (Moderna), and one each evaluating CoronaVac (Sinovac), BBIBP-CorV (Sinopharm), Ad5-nCoV (CanSino), and ZyCoV-D (Zydus Cadila) (Table 1). 

Immunogenicity was described as seroconversion of neutralising antibodies (4 studies), geometric mean titre (GMT) after primary vaccine series (8 studies), or GMT ratio comparing immune response in children/adolescents to that in the young adult population (4 studies). Seroconversion of neutralising antibodies in vaccinated individuals who received ZyCoV-D (Zydus Cadila), CoronaVac (Sinovac), Ad5-nCoV (CanSino), and mRNA-1273 (Moderna) vaccines ranged from 88 to 100 percent (Figure 2A). The BNT162b2 (Pfizer–BioNTech) and mRNA-1273 (Moderna) vaccines showed non-inferiority of immune response in children/adolescents compared with the young adult population (16 to 25 years old), with GMT ratios between the two groups ranging from 1.04 to 1.76 (Figure 2B).

Efficacy against confirmed COVID-19 was evaluated in five clinical trials. Efficacy of the mRNA-1273 (Moderna) vaccine against symptomatic COVID-19 within 14 days after completion of the primary series was 93.3% (95% CI 47.9–99.9). These studies were performed during and prior to the Delta variant circulation. Efficacy of the BNT162b2 (Pfizer-BioNTech) vaccine against confirmed COVID-19 was described in three separate studies as 100% (95% CI 58.2–100) within 7 days after the second dose and 90.7% (95% CI 67.4–98.3) to 100% (95% CI 75.3–100) at least 7 days after the second dose (Figure 2C). Most of these studies were completed before the Delta variant circulation. Table 1 shows the dates of trial conduction.

### 3.5. Vaccine Effectiveness

Effectiveness was measured against non-critical COVID-19 infection/outpatient encounters, hospitalisation, and severe COVID-19 infection (MIS-C, COVID-19 pneumonia, or ICU admission). VE was described during the Delta- and Omicron-predominant phases of the pandemic for children (5–11 years) and adolescents (12 years and above).

Vaccine effectiveness of the BNT162b2 (Pfizer-BioNTech) primary vaccine series against non-critical SARS-CoV-2 infection was evaluated in 10 papers. In adolescents during the Delta variant circulation, VE ranged from 85% (95% CI 81–89) to 93% (95% CI 76–98). VE was lower during the Omicron variant circulation among this age group, ranging from 20% (95% CI −25–49) to 62% (95% CI 30–79). In the 5-to-11-year age group, VE during the Omicron variant circulation ranged from 47% (95% CI 32–59) to 51% (95% CI 30–65). Overall VE during the Delta and Omicron periods combined was described as 46% (95% CI 24–61) in the 5- to 11-year-olds, 83% (95% CI 80–85) in the 12- to 15-year-olds, and 76% (95% CI 71–80) in the 16- to 17-year-olds. 

Vaccine effectiveness of the BNT162b2 (Pfizer-BioNTech) primary vaccine series against hospitalisation was evaluated in five studies (Figure 2D). Vaccine effectiveness against hospitalisation varied among age groups and differed during the Delta- and Omicron-predominant periods. For adolescents, VE against hospitalisation was 92% (95% CI 89–95) during Delta, compared with 40% (95% CI 9–60) during Omicron. VE against hospitalisation was higher among older children compared with younger: 94% in the 16–17 age group (95% CI 87–97) and 92% in the 12–15 age group (95% CI 79–97), compared with 74% in the 5–11 age group (95% CI 35–95). 

Vaccine effectiveness of the BNT162b2 (Pfizer-BioNTech) vaccine against severe illness requiring non-invasive and invasive mechanical ventilation, vasoactive infusions, or extracorporeal membrane oxygenation during hospital stay among adolescents was 96% (95% CI 90–98) during Delta and 79% (95% CI 51–91) during Omicron circulation [41]. VE against MIS-C was 91% (95% CI 78–97) [18]. For the “China-made” COVID-19 vaccines, including inactivated, adenovirus-vectored, and recombinant-subunit vaccines, the odds ratio of COVID-19 pneumonia was 0.42 (95% CI 0.31–0.57) during Delta and 1.44 (95% CI 0.32–6.37) during Omicron circulation.

### 3.6. Expert Survey

We received a total of 118 survey responses from vaccination experts representing 55 different countries. At the time of survey analysis on 15 June 2022, there were 38 COVID-19 vaccine types approved for use globally [54], including mRNA, protein subunit, inactivated, and non-replicating viral vector vaccines. Survey respondents reported that twelve vaccine types were being used for individuals under 18 years of age. The mRNA vaccines, manufactured by Pfizer and Moderna, were used in most countries, followed by vaccines produced by Sinovac, AstraZeneca, Sinopharm, Novavax, and Janssen. Countries also reported use of vaccines manufactured by Corbevax, Covaxin, Convidecia, Covavax, and Sputnik in children.

The reported age eligibility for vaccination ranged from >3 years old to >18 years old. Countries offering vaccination for children as young as three years old at the time of the survey included Argentina, Chile, and Colombia. Most countries allowed all children who met age eligibility to be vaccinated. Based on the survey, two countries only allowed children with comorbidities to be vaccinated.

The reported vaccination rate of age-eligible children having received at least one vaccine dose varied greatly across countries, ranging from <1% to 98%. New Zealand reported the highest vaccination rate, followed by Chile and Argentina. For countries that did not provide a vaccination rate in their survey response, our team attempted to look up the data on government and public health websites [54,55,56,57,58,59] and included it on the vaccination rate map (Figure 3b) if data were available.

Ten countries reported vaccine mandates for children for one or more of the following: travel, school attendance, and sports participation. Three countries reported having vaccine mandates earlier in the pandemic that no longer exist. Of the ten countries reporting vaccine mandates, one allowed religious exemptions and four allowed medical exemptions. None accepted personal belief exemptions.

When asked about public opinion towards vaccination policies for children, the majority of respondents stated, in their opinion, that there were “mixed opinions” in their country. Approximately one-third reported that they felt that the public was satisfied with current mandates, while 11% reported that the public desired fewer mandates, and 4% reported that the public desired more mandates. 

The survey inquired about respondents’ thoughts on public attitudes towards COVID-19 vaccination in children, on a scale of negative 10 (mostly negative) to positive 10 (mostly positive). Figure 3a depicts the scaled public attitudes as an average of all respondents by country. When comparing reported public attitudes with vaccination rates, we found a small but significant correlation between having a more positive public attitude towards vaccination and having a higher vaccination rate (R^2^ = 0.18, *p* = 0.02).

Reasons for vaccine hesitancy in children as observed by experts were described in five categories: safety concerns, feeling that immunisation is unnecessary, mistrust, religious beliefs, and political objections. The majority of respondents cited “safety” and “unnecessary” as reasons for vaccine hesitancy. Regarding safety, survey respondents stated that public perception of the vaccine included being too new and developed too quickly, unknown safety profile or long-term effects, adverse side effects, and risks outweighing benefits. Respondents also cited public belief that the vaccine was unnecessary given mild symptoms similar to the common cold, declining case numbers in respective countries, children having already been infected by COVID-19, and the disease being less severe in children. Some respondents stated that they felt the public has mistrust of the vaccines and disbelief in the pandemic, with social media and anti-vaccination groups playing a large role in the dissemination of information. Regarding religious beliefs as a reason for vaccine hesitancy, some sub-Saharan African countries feared that the vaccines may be related to “666” signifying “The Beast” in Christianity. There was also concern that vaccine ingredients may not be Halal or permissible in countries with Muslim-predominant citizens. Political concerns included government objection from conservative groups in various countries and the government undermining vaccination initiatives in parts of South America. The reasons for COVID-19 vaccine hesitancy in children reported by survey respondents are shown in the Figure 4.

## 4. Discussion

Vaccination against SARS-CoV-2 in children has a building evidence base of safety, immunogenicity, efficacy, and effectiveness. Side effects are common after vaccination but generally mild and self-limiting. Previous reviews of vaccine safety in both children and adults found similar results [60,61,62]. The nine primary vaccine trials studying six different vaccine types demonstrated favourable immunogenicity and efficacy among the paediatric population, either through seroconversion of neutralising antibodies, non-inferiority of GMT compared with young adults, or prevention of COVID-19 infection after completing the primary vaccine series. The BNT162b2 (Pfizer-BioNTech) vaccine was found to be effective in preventing hospitalisation and critical infection among children and adolescents when evaluated in six studies including over 50,000 participants. Vaccine effectiveness varied during different variant-predominant periods; there was decreased effectiveness seen during the Omicron-variant-predominant period compared with the Delta-variant-predominant period [41]. 

Despite this growing body of evidence, the perceptions of COVID-19 vaccine safety is mixed in both the public and professional spheres. Our survey data provides a snapshot in time of the global status of COVID-19 vaccines in children, showing great variation in vaccination policies, rates, and attitudes. Critically, two years into the pandemic and over a year since COVID-19 vaccines were licensed for adolescents, gaps in knowledge continue, even among vaccine experts, regarding novel vaccine types and safety data in children. The WHO has highlighted evidence in children and adolescents as being a critical knowledge gap [63]. Children and adolescents should be included in large-scale clinical trials, with results published rapidly and shared openly. In countries where vaccination of children and adolescents is already being undertaken, clinicians, parents and children should be encouraged to report side effects to national reporting systems so that trends and critical events can be monitored. 

Globally, national policies relating to the vaccination of children and adolescents are mixed and occur for a variety of reasons. The WHO had set a target of 70% for global vaccination coverage by mid-2022, of which only 58 of 194 member states had accomplished [64]. The majority of vaccine doses have been administered in high- and upper middle-income countries, with primary vaccination and booster vaccination being highly regressive and inequitable [65]. Adolescents and children in low-income settings are likely to be a low priority for vaccination and may wait much longer than their peers in high-income countries before they are offered vaccination. The choice of vaccine available is likely to limit vaccination of adolescents or children. The COVID-19 Vaccines Global Access (COVAX) facility initially purchased large supplies of AstraZeneca, Janssen, and Novavax [65], for which efficacy and safety data in children and adolescents is more limited than that of other vaccine types or suppliers. Our survey results highlighted that in some settings with limited access to vaccines, children and adolescents may pragmatically receive what is available when faced with no alternative rather than what is known to be safe. This is another facet of COVID-19 vaccine inequity that requires global debate.

### Limitations

Our survey was sent to members of vaccination and public health initiatives which introduces a selection bias. Survey answers may reflect survey respondents’ perceptions rather than overall public beliefs. With frequent policy updates, a recall bias is also possible. The responses also show only a short snapshot in time in a rapidly changing field. 

Our review of the literature also has limitations. The review was undertaken in a systematic way; however, not all steps for a strict systematic review were undertaken, such as protocol registration and formal risk of bias scoring, in order to increase the speed at which results could be analysed. Authors discussed studies, including study attributes and risk of bias, at regular meetings. The results presented are for published studies at the time of our review. There are over 200 clinical trials involving children and COVID-19 vaccination registered on clinicaltrials.gov, and the vast majority are yet to share results [66]. Efficacy data from mRNA vaccine clinical trials for children aged six months to four years are expected to be released soon. Ongoing effectiveness studies are needed with new circulating variants and with new multivalent vaccines. A living systematic review of vaccine safety, immunogenicity, and efficacy is required to ensure that the best evidence can be shared with patients, parents, and health care professionals. 

We found that age-disaggregated data could not be extracted from some studies, and there was no response from authors for this data. In order to improve vaccine confidence, researchers and pharmaceutical companies should make their data-sets available for analysis. Data pertaining to adverse events were also not uniformly reported across studies. 

The Brighton Collaboration, alongside the Coalition for Epidemic Preparedness (CEPI), has developed the Safety Platform for Emergency vACcines (SPEAC) [67]. This project has developed a list of Adverse Events of Special Interest (AESI) associated with COVID-19 vaccines, endorsed by the WHO Global Advisory Committee on Vaccine Safety. Researchers should be encouraged to present data with standardised case definitions in a uniform way that enables rapid analysis of data. 

## 5. Conclusions

In this review, we have found evidence that vaccination against COVID-19 in children is safe, efficacious, and effective. Perceptions surrounding vaccines are mixed and global policies mirror this whilst also reflecting global vaccine inequity. Overall, the combined evidence from both the review of the literature and the survey highlights the need for further data on both the safety and effectiveness of COVID-19 vaccinations in children. 

## Figures and Tables

**Figure 1 vaccines-11-00078-f001:**
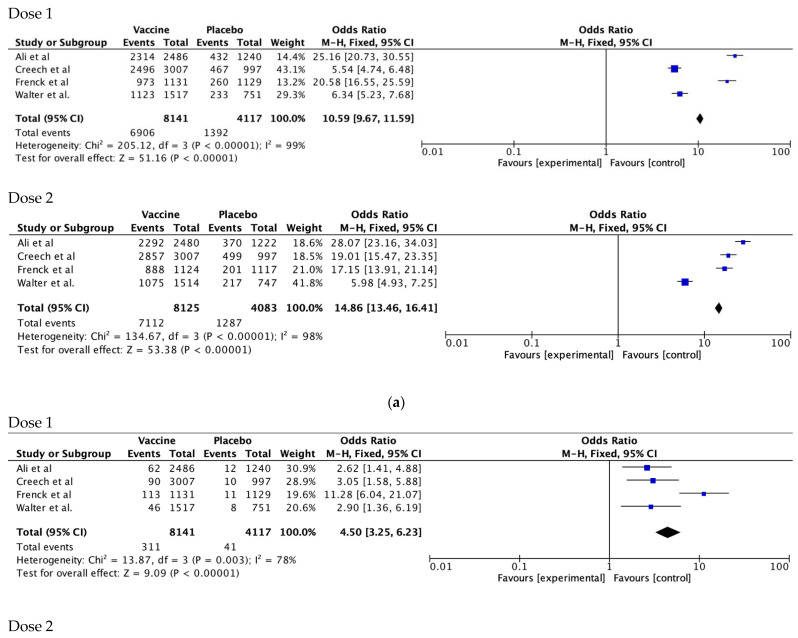
(**a**) Forest plots comparing injection-site pain in participants receiving mRNA vaccines compared with placebos after Dose 1 and Dose 2 [24,26,31,33]. (**b**) Forest plots comparing fever in participants receiving mRNA vaccines compared with placebos after Dose 1 and Dose 2 [24,26,31,33].

**Figure 2 vaccines-11-00078-f002:**
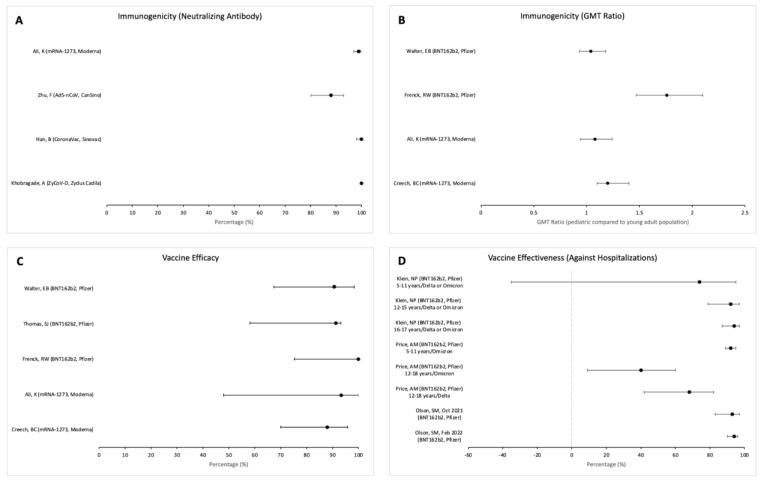
Forest plots describing (**A**) percent of the paediatric study population showing seroconversion of neutralising antibodies after the primary COVID-19 vaccine series [25,26,29,32], (**B**) geometric mean titre (GMT) ratio of paediatric study population compared with young adult population [24,26,31,33], (**C**) vaccine efficacy in paediatric COVID-19 vaccine clinical trials [24,26,27,31,33], and (**D**) vaccine effectiveness against COVID-19-related hospitalizations [35,40,41].

**Figure 3 vaccines-11-00078-f003:**
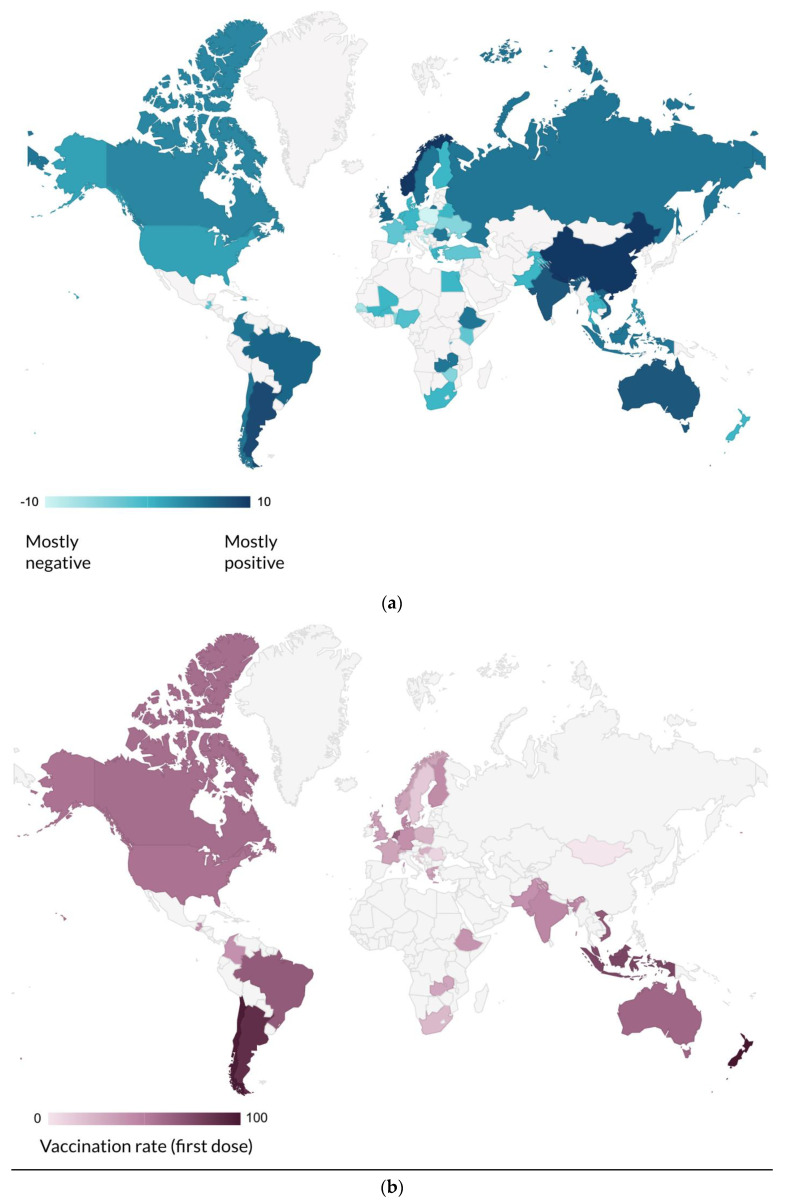
(**a**) Global maps depicting reported public attitude towards vaccinating children against COVID-19 and vaccination rates, April–June, 2022. (**b**) The reported vaccination rate of age-eligible children having received at least one vaccine dose.

**Figure 4 vaccines-11-00078-f004:**
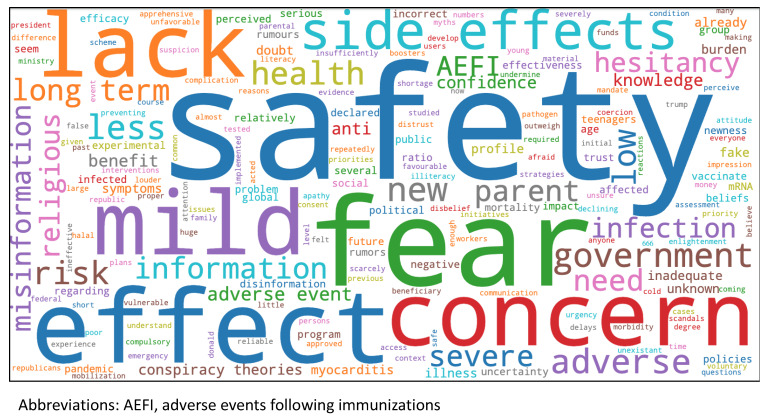
Reasons for COVID-19 hesitancy in children reported by a survey of global vaccination experts are shown in a word cloud where more frequent responses are represented by a larger word size.

**Table 1 vaccines-11-00078-t001:** Studies included in the safety and efficacy review of the literature.

First Author	Country	Publication Date	Vaccine Type	Study Type	Study Phase	Age Group (Years)	Population Size	Study Enrolment Dates	Immunogenicity: Seroconversion of Neutralizing Antibody (95% CI) *28 Days after Primary Series*	Immunogenicity: Geometric Mean Titre or GMT (95% CI) *28 Days after Primary* *Series*	Immunogenicity: GMT Ratio Compared to Young Adult Group (95% CI)	Efficacy (95% CI)
Frenck, RW [24]	USA	May 2021	BNT162b2 (Pfizer-BioNTech)	Randomized placebo- controlled trial (RPCT)	3	12 to 15	2260	Oct 2020– Jan 2021		1239.5 (1095.5 to 1402.5)	1.76 (1.47 to 2.10) compared with 16–25-year-olds	Against confirmed COVID-19 was 100% (75.3 to 100) at least 7 days after second dose
Han, B [25]	China	June 2021	CoronaVac (Sinovac)	RPCT	1–2	3 to 17	552	Oct 2020– Dec 2020	100.0% (98.0 to 100.0) in 3.0 ug group, phase 2 trial	142.2 (124.7 to 162.1) at 3.0 ug		
Ali, K [26]	USA	Aug 2021	mRNA-1273 (Moderna)	RPCT	2–3	12 to 17	3732	Dec 2020– Feb 2021	98.8% (97.0 to 99.7)	1401.7 (1276.3 to 1539.4)	1.08 (0.94 to 1.24) compared with 18–25-year-olds	Against symptomatic infection was 93.3% (47.9 to 99.9) within 14 days after second dose
Thomas, SJ [27]	USA, Argentina, Brazil, South Africa, Germany, Turkey	Sept 2021	BNT162b2 (Pfizer-BioNTech)	RPCT	1–2–3	16 to 17 (stratified)	683	July 2020– Oct 2020				Against confirmed COVID-19 was 100% (58.2 to 100) within 7 days after second dose
Xia, S [28]	China	Sept 2021	BBIBP-CorV (Sinopharm)	RPCT	1–2	3 to 17	1255	Aug 2020– Sept 2020		180.2 (163.4 to 198.8) at 4 μg		
Zhu, F [29]	China	Sept 2021	Ad5-nCoV (CanSino)	RPCT	2b	6 to 17 (stratified)	150	Sept 2020– Nov 2020	88.0% (80.2 to 93.0)	96.6 (76.8 to 121.4)		
Vadrevu et al. [30]	India	Dec 2021	BBV152	Open label	2–3	2 to 18	526	May 2021– July 2021				
Walter, EB [31]	USA, Spain Finland, Poland	Jan 2022	BNT162b2 (Pfizer-BioNTech)	RPCT	1–2–3	5 to 11	2334	Mar 2021– Jun 2021		1197.6 (1106.1 to 1296.6) at 10 ug	1.04 (0.93 to 1.18) compared with 16–25-year-olds	Against confirmed COVID-19 was 90.7% (67.4–98.3) at least 7 days after second dose
Khobragade, A [32]	India	April 2022	ZyCoV-D (Zydus Cadila)	RPCT	3	12 to 17 (stratified)	935	Jan 2021– June 2021	100%	2083 (EU)		
Creech, BC [33]	USA/Canada	May 2022	mRNA-1273 (Moderna)	RPCT	2–3	6 to 11	4016	March 2021– August 2021 (Delta)		1610 (1457 to 1780) at 50 μg	1.2 (1.1 to 1.4) compared with 18–25-year-olds	Against confirmed COVID-19 was 88.0% (70.0 to 95.8) at least 14 days after the dose

**Table 2 vaccines-11-00078-t002:** Studies included in the vaccine effectiveness review of the literature.

First Author	Country	Publication Date	Vaccine Type	Study Type	Age Group (Years)	Population Size	Study Enrolment Dates (Variant)	Effectiveness against Non-Critical COVID-19 Infection or Outpatient Encounters VE% (95% CI) *≥14 Days after Primary Series unless* *Otherwise Noted*	Effectiveness Against Hospitalization VE% (95% CI) *≥14 Days after Primary Series*	Effectiveness Against Severe COVID-19 or ICU Admission
Reis, BY [34]	USA	Oct 2021	BNT162b2 (Pfizer-BioNTech)	Retrospective case control	12 to 18	184,905	June 2021–Sept 2021 (Delta)	93 (88 to 97) against symptomatic infection, days 7 to 21 after second dose		
								90 (88 to 92) against documented infection, days 7 to 21 after second dose		
Olson, SM [35]	USA	Oct 2021	BNT162b2 (Pfizer-BioNTech)	Case-control study	12 to 18	572	June 2021–Sept 2021 (Delta)		93 (83 to 97)	No vaccinated adolescents were admitted to the ICU
Glatman-Freedman, A [36]	Israel	Nov 2021	BNT162b2 (Pfizer-BioNTech)	Retrospective cohort study	12 to 15	All 12–15 years old in Israel excluding those with positive COVID-19 PCR	July 2021–Aug 2021 (Delta)	88.2 (85.0 to 90.7)	No vaccinated adolescents who became SARS-CoV-2-positive were hospitalized 1–28 days after primary vaccine series	
Choe, YJ [17]	South Korea	Dec 2021	BNT162b2 (Pfizer-BioNTech)	Retrospective cohort study	16 to 18	1,299,965	July 2021–Nov 2021 (Delta)	Against asymptomatic or symptomatic COVID infection:		
								99.1 (98.5 to 99.5)		
Lutrick, K [37]	USA	Dec 2021	BNT162b2 (Pfizer-BioNTech)	Prospective cohort study	12 to 17	243	July 2021–Dec 2021 (Delta)	94 (83 to 98)		
Zambrano, LD [18]	USA	Jan 2022	BNT162b2 (Pfizer-BioNTech)	Retrospective case control	12 to 18	117	July 2021–Dec 2021 (Delta)			91 (78 to 97) against MIS-C
										No vaccinated adolescent with MIS-C required life support
										One patient (20%) required ICU admission
Lin, DY [38]	USA	Jan 2022	BNT162b2 (Pfizer-BioNTech)	Retrospective review study	12–17 (stratified)	806,634	Dec 2020–Sept 2021 (Delta)	95.2 (94.5 to 95.7) one month after first dose		
Olson, SM [35]	USA	Feb 2022	BNT162b2 (Pfizer-BioNTech)	Retrospective case control	12 to 18	1222	July 2021–Oct 2021 (Delta)		94 (90 to 96)	98 (93 to 99)
Oliveira, CR [39]	USA	March 2022	BNT162b2 (Pfizer-BioNTech)	Retrospective case control	12 to 18	542	June 2021–Aug 2021 (Delta)	91 (80 to 96) symptomatic 93 (81 to 97) asymptomatic 94 (75 to 98) during Delta		
Klein, NP [40]	USA	March 2022	BNT162b2 (Pfizer-BioNTech)	Retrospective review study	5 to 17	40,916 urgent care, emergency department, or hospitalization encounters	Apr 2021–Jan 2022	46 (24 to 61) in 5–11 year-olds; 51 (30 to 65) during Omicron	74 (−35 to 95) in 5–11 year-olds during Delta or Omicron	
							(Delta/Omicron)	83 (80 to 85) in 12–15 year-olds; 92 (89 to 94) during Delta; 45 (30 to 57) during Omicron	92 (79 to 97) in 12–15 year-olds during Delta or Omicron	
								76 (71 to 80) in 16–17 year-olds; 85 (81 to 89) during Delta; 34 (8 to 53) during Omicron	94 (87 to 97) in 16–17 year-olds during Delta or Omicron	
Price, AM [41]	USA	March 2022	BNT162b2 (Pfizer-BioNTech)	Case-control study	5 to 18	2812	July 2021–Feb 2022 (Delta/Omicron)	91 (86 to 94) in 12–18 year-olds during Delta	92 (89 to 95) in 12–18 year-olds during Delta	96 (90 to 98) in 12–18 year-olds during Delta
								20 (−25 to 49) in 12–18 year-olds during Omicron	40 (9 to 60) in 12–18 year-olds during Omicron	79 (51 to 91) in 12–18 year-olds during Omicron
									68 (42 to 82) in 5–11 year-olds during Omicron	
Fowlkes, AL [42]	USA	March 2022	BNT162b2 (Pfizer-BioNTech)	Prospective cohort study	5 to 15	1364	July 2021–Feb 2022 (Delta/Omicron)	Against asymptomatic or symptomatic COVID:		
								47 (32 to 59) in 5–11 year-olds during Omicron		
								93 (76 to 98) in 12–15 year olds during Delta		
								62 (30 to 79) in 12–15 year olds during Omicron		
Li, M [16]	China	April 2022	“China-made COVID-19 vaccines” (inactivated, adenovirus-vectored, recombinant-subunit vaccine)	Retrospective, descriptive analysis	3 to 17 (stratified)	1196	May 2021–Feb 2022 (Delta/Omicron)			Odds Ratio of COVID pneumonia: 0.42 (0.31–0.57) during Delta
										1.44 (0.32–6.37) during Omicron
Cohen-Stavi, CJ [43]	Israel	July 2022	BNT162b2 (Pfizer-BioNTech)	Observational cohort study	5 to 11	556,808	Nov 2021–Jan 2022 (Omicron)	Against documented infection: 51 (39 to 61)		
								68 (43 to 84) in 5–6 year-olds		
								56 (41 to 68) in 7–9 year-olds		
								38 (18 to 53) in 10–11 year-olds		
								Against symptomatic infection: 48 (29 to 63)		
								69 (30 to 91) in 5–6 year-olds		
								49 (6 to 76) in 7–9 year-olds		
								36 (0 to 61) in 10–11 year-olds		
								At 7 to 21 days after primary vaccine series		

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
