# Peer review of "Vaccinating Children against SARS-CoV-2: A Literature Review and Survey of International Experts to Assess Safety, Efficacy and Perceptions of Vaccine Use in Children"

_vaccines, 2022, doi:10.3390/vaccines11010078_

Round 1

Reviewer 1 Report

Thank you for the chance to review this paper

I would appreciate you addressing the following:

Under Introduction:

Line 40-46: there are new data on long covid, please update the numbers. Similarly, whereas MIS-C and long covid are important aspects of covid infection in children, so are hospitalisations. Please include a reference to the hospitalisation burden from covid in pediatrics

Line 51: whereas lockdown measures were significant at the height of the pandemic, they are less likely to influence vaccination intention at this stage

Line 167 mentions the age group 16-25, yet in line 145, the age range was described between 6 months and 18 years. Please correct this discrepency

Lines 233-238, in the description of the efficacy studies, please clarify during which variant period they were conducted

Lines 318-334: these describe the expert survey responses. This is a heavily biased  part of the study, as it is virtually impossible to accurately conclude public perception based on surveying experts (for example, not only is recall bias an issue, but also personal opinion heavily colours their responses). Please use language that actually describes the findings: e.g: Reasons for vaccine hesitancy should be described as: reasons for vaccine hesitancy as observed by the experts, are: etc etc. Similarly, in lines 326-33 the authors describe what is likely data obtained from a free text/comments portion of the survey. What methodology was used to analyse these? One assumes this portion utilized qualitative research methids: if so, which and how was it done? It seems to me that the "reasons" identified in lines 330-332 were the high impact findings, but not sure whether they also identified as important by frequency in the responses.

Line 404: the authors mention mixed methods review, but I do not see any mention of the qualitative methods or what type of mixed methods were employed. Please address this: a survey that has quantitative data and some comments options does not necessarily qualify as mixed methods research.

Reviewer 2 Report

This paper aimed to summarise the evidence for a range of outcomes of Covid-19 vaccination in children. While on the one hand they conclude that they found evidence that vaccination in children is safe, efficacious and effective, they also conclude their study highlights the need for further data. These apparent contradictory statements are not further discussed, but could be related to the methodological weaknesses of the current study, which makes is scientific and practical value limited. Furthermore, while the justification of the study is introduced as the potential risks of MIS-C and long covid following infection, as well as the lack of consensus on the need for vaccination of children, this study does not respond to this. This would require a study in which the assessment of safety, effectiveness and acceptability of vaccination is balanced against the evidence for benefits, which this study does not address. Therefore, either the background/introduction, or the remainder of the paper, needs to be overhauled.

The study consisted of three different methods (a literature review, a search of safety reports, and a survey), but what is lacking is a comprehensive framework justifying this combination of methods. Also, more information needs to be provided to assess validity and quality. Eg, for the literature review, it should be clarified who searched them (2 independent reviewers, with a third reviewer in case of discrepancies?), provide a flow chart to summarise the selection process, explain why 2 searches were done, why one article was added after the second search, rather than ensuring a systematic search at a single final date, etc. For the safety databases: explain the selection of countries, why two searches, implications for not including any LMIC countries in that search (while LMIC data were included in the literature review), etc. For the survey, no information is provided on the positions or backgrounds (including geographical location) of people invited, nor of people who responded; why was the survey send three times, (how) were the survey questions developed, tested and validated, etc. In the limitations the authors state that due to lack of time, they could not follow validated methodologies. However, results of poor quality have little added value, so this argument seems very odd, as the rush is not clarified.

Other statements are confusing also. What does it mean that articles were excluded if they could not be translated into English or Spanish? How many articles included were translated and how (translator, machine translation, ..) ; which articles could not be translated? It is stated that 9 RCTs were included, but in Table 1, 10 studies are mentioned (although no reference number provided)? An AE is reported for a participant aged 16-25 years: how is this included in an assessment of paediatric vaccines, and none of the studies mentioned in table 1 has this age category reported? In table 1, 3 studies are described as multinational: please provide details how many children where.

Proper labels for the figures and tables are lacking.

For the results, please explain why no summary statistics were provided, and clarify which paper was the one added ‘at the discretion of the authors’ only. The correlation between a reported positive public attitude and higher uptake is small, which is acknowledged when first reported, but overstated in the discussion and abstract. Also in view of the serious methodological limitations of the survey, this should be modified. 

Reviewer 3 Report

Must be revised properly and I also find that some contents may address plagiarism. Authors are Requested to address the plagiarism issue. Please also revise introduction and discussion with relevant Latest references. 

Author Response

Dear reviewer 3,

Thank you for taking the time to review our paper. We hope to have addressed your constructive feedback. You can see our responses in blue underneath the points that you have raised.

Must be revised properly and I also find that some contents may address plagiarism. Authors are Requested to address the plagiarism issue. Please also revise introduction and discussion with relevant Latest references. 

Thank you for your comments. We did not plagiarise any text and have run the manuscript through anti-plagarism software.

The score is <1% which we feel highlights the originality of our text.

Reviewer 4 Report

This is a useful study regarding Vaccination of children against SARS-CoV-2. However, some parts of the manuscript should be improved before considering it for publication.

The tables and figures are not clear for the reader and it should be replaced with high resolution one.

the firsdt table in the manuscript has no title and the content is not clear, I think this table should be presented in a more simple manner.

The discussion section needs to be improved and all data should be comprhinsivly discussed. 

Author Response

Dear reviewer 4,

Thank you for taking the time to review our paper. We hope to have addressed your constructive feedback. You can see our responses in blue underneath the points that you have raised.

This is a useful study regarding Vaccination of children against SARS-CoV-2. However, some parts of the manuscript should be improved before considering it for publication.

The tables and figures are not clear for the reader and it should be replaced with high resolution one.

We have included higher resolution figures and table. We will be happy to supply separate files with high resolution images for inclusion if the paper is accepted.

the firsdt table in the manuscript has no title and the content is not clear, I think this table should be presented in a more simple manner.

Thank you, we have added a title and improved the table as you suggest

The discussion section needs to be improved and all data should be comprhinsivly discussed. 

We have included additional information in the discussion (see comments to queries from other reviewers which highlighted a similar need for further changes to the discussion).

Round 2

Reviewer 2 Report

Thank you for modifying the manuscript in line with some of my suggestions, although it was a bit surprising to read in the 4th para of the reply to my first comment that MIC-S and long covid were considered associated with vaccination (fortunately, this is not formulated as such in the draft manuscript). 

Other discrepancies remain also. Eg, it is still unclear to what extent a proper (final) search was done in September 2022 (also, the text mentions one paper was added, at the discretion of senior author, line 148, the reply mentions two additional papers were identified by September 2022). The reference for this additional paper (24) is incomplete.

As the authors already indicate now themselves, combining a systematic review with a survey among a selected group of self-acclaimed experts, is not a strong design. Both parts remain with limitations, combining them does not create synergy, but distracts from any main message, combining merged data analysis of RCTs, with summarised, non-validated opinions, without a theoretical phramework, as pointed out before in my second comment.  It remains highly recommend NOT to combine these. Also the review itself would be more coherent if either the focus would be on the outcome of RCTs (short term relatively common adverse effects, immunogenicity, efficacy), or focus on either of these three aspects in depth, assessing both RCT and 'real life' longer term/rare events systematically (be it focused on safety, immune waning/boosting, or effectiveness). Now we have a review of 10 or 11 RCTs, with some information on a incomplete inventory of safety databases and some effectiveness studies. A little bit of everything adds little.

Author Response

Dear Editors,

We feel that reviewers overwhelmingly consider that we have addressed their comments. We respectfully disagree with reviewer 2 as we feel that all elements of the study add to the paucity of data on safety and immunogenicity in children. We undertook a systematic review of the literature with follow up in the main scientific database and extracted data from all publicly available national safety databases. If the reviewer is able to suggest specific RCT or databases that they feel we have missed, we are more than happy to include them. Please see our point-by-point responses below to reviewer 2.

Thank you for modifying the manuscript in line with some of my suggestions, although it was a bit surprising to read in the 4th para of the reply to my first comment that MIC-S and long covid were considered associated with vaccination (fortunately, this is not formulated as such in the draft manuscript). 

Vaccination is associated with protection from MIS-C. We included this reference (ref. #18) in the introduction. Vaccination is likely to also protect against long COVID given its effectiveness against COVID disease. However, there have not been studies published on this question at this time.

Other discrepancies remain also. Eg, it is still unclear to what extent a proper (final) search was done in September 2022 (also, the text mentions one paper was added, at the discretion of senior author, line 148, the reply mentions two additional papers were identified by September 2022). The reference for this additional paper (24) is incomplete.

We thank the reviewer for their helpful comments. As we note in the methods section, and replied in the previous rebuttal, due to the number and rapidly moving field of COVID19, we were unable to undertake a full systematic review following our original searches. However, a search of the NCBI database revealed an additional 2 papers that met eligibility to be included. One of those included safety data (line 148, reference 33). The second reported vaccine effectiveness data (ref 56).

We added the full reference information for ref 24 which is now ref. 33.

As the authors already indicate now themselves, combining a systematic review with a survey among a selected group of self-acclaimed experts, is not a strong design. Both parts remain with limitations, combining them does not create synergy, but distracts from any main message, combining merged data analysis of RCTs, with summarised, non-validated opinions, without a theoretical phramework, as pointed out before in my second comment.  It remains highly recommend NOT to combine these.

We respectfully disagree with the reviewer. As we point out, the purpose of the survey was to understand vaccine coverage in children as many countries do not have publicly available national databases that report this. The experts all work in the field of national public health and immunisation and therefore are best placed to inform coverage rates in the absence of national data. We therefore feel that this element of the survey adds value to the trial and pharmacovigilance databases we reviewed. If the reviewer is able to suggest alternative sources of national data on the scale we report, we would be happy to include that instead.

Also the review itself would be more coherent if either the focus would be on the outcome of RCTs (short term relatively common adverse effects, immunogenicity, efficacy), or focus on either of these three aspects in depth, assessing both RCT and 'real life' longer term/rare events systematically (be it focused on safety, immune waning/boosting, or effectiveness). Now we have a review of 10 or 11 RCTs, with some information on a incomplete inventory of safety databases and some effectiveness studies. A little bit of everything adds little.

We feel that including both RCT and national safety databases does add value as it provides the opportunity to report both common and rare adverse events, as you point out. We also reported on all RCTs which were published at the time of our review. We undertook a thorough search of all national pharmacovigilance databases that were publicly available and report on all of these. Finally, we reported on all vaccine effectiveness studies that included children published at the time of our review.  If the reviewer feels that there are any missing, we would be happy to investigate this further.

Reviewer 3 Report

No comments

Author Response

There were no specific comments from this reviewer. We will be happy to address anything which is raised in further detail. 

Reviewer 4 Report

The authors revised the manuscript appropriately.

Author Response

There are no comments from reviewer 4. We will be happy to address any if they arise.